# Biodegradable Thermoplastic Starch/Polycaprolactone Blends with Co-Continuous Morphology Suitable for Local Release of Antibiotics

**DOI:** 10.3390/ma15031101

**Published:** 2022-01-30

**Authors:** Veronika Gajdosova, Beata Strachota, Adam Strachota, Danuse Michalkova, Sabina Krejcikova, Petr Fulin, Otakar Nyc, Adam Brinek, Marek Zemek, Miroslav Slouf

**Affiliations:** 1Institute of Macromolecular Chemistry of the Czech Academy of Sciences, Heyrovsky Sq. 2, Prague 6, 162 06 Prague, Czech Republic; gajdosova@imc.cas.cz (V.G.); beata@imc.cas.cz (B.S.); strachota@imc.cas.cz (A.S.); DanuseMichalkova@seznam.cz (D.M.); krejcikova@imc.cas.cz (S.K.); 21st Orthopedic Clinic, 1st Faculty of Medicine, Charles University, Motol University Hospital, V Uvalu 84, Prague 5, 150 06 Prague, Czech Republic; petrfulin@gmail.com; 3Department of Medical Microbiology, 2nd Faculty of Medicine, Charles University, Motol University Hospital, V Uvalu 84, Prague 5, 150 06 Prague, Czech Republic; otakar.nyc@fnmotol.cz; 4CEITEC—Central European Institute of Technology, Brno University of Technology, Purkynova 656/123, 612 00 Brno, Czech Republic; Adam.Brinek@ceitec.vutbr.cz (A.B.); Marek.Zemek@ceitec.vutbr.cz (M.Z.)

**Keywords:** thermoplastic starch, poly(ε-caprolactone), polymer blends, micromechanical properties, microindentation, structure–properties relations

## Abstract

We report a reproducible preparation and characterization of highly homogeneous thermoplastic starch/pol(ε-caprolactone) blends (TPS/PCL) with a minimal thermomechanical degradation and co-continuous morphology. These materials would be suitable for biomedical applications, specifically for the local release of antibiotics (ATB) from the TPS phase. The TPS/PCL blends were prepared in the whole concentration range. In agreement with theoretical predictions based on component viscosities, the co-continuous morphology was found for TPS/PCL blends with a composition of 70/30 wt.%. The minimal thermomechanical degradation of the blends was achieved by an optimization of the processing conditions and by keeping processing temperatures as low as possible, because higher temperatures might damage ATB in the final application. The blends’ homogeneity was verified by scanning electron microscopy. The co-continuous morphology was confirmed by submicron-computed tomography. The mechanical performance of the blends was characterized in both microscale (by an instrumented microindentation hardness testing; MHI) and macroscale (by dynamic thermomechanical analysis; DMTA). The elastic moduli of TPS increased ca four times in the TPS/PCL (70/30) blend. The correlations between elastic moduli measured by MHI and DMTA were very strong, which implied that, in the future studies, it would be possible to use just micromechanical testing that does not require large specimens.

## 1. Introduction

Due to environmental and sustainability issues, biodegradable polymers have attracted more and more attention [1,2]. The most widely used biodegradable polymers are starch and poly (lactic acid) [3]. Starch is popular due to its natural origin, abundance connected with a low cost, excellent biocompatibility, and complete biodegradability [4]. Natural starch is a white, non-plastic powder which decomposes before melting [4,5]. This natural powder must be plasticized so that it can be treated by classical polymer processing techniques such as melt-mixing or compression molding [6,7,8,9]. The starch plasticization (or gelatinization) consists of mixing starch powder with low molecular weight compounds such as water, glycerol, citric acid, etc. [10,11,12], and results in thermoplastic starch (TPS). Two main ways of starch plasticization are solution casting (SC; [13,14] and melt-mixing (MM; [6,10,15]). A series of our studies [16,17,18] proved that highly homogenous TPS (i.e., material without microscale inhomogeneities, suitable not only for technical, but also for biomedical applications) is best obtained by a two-step preparation procedure, which comprises SC followed by MM.

In most applications, TPS is used in the form of blends with other polymers [19]. The reason is that neat TPS is rather soft, very sensitive to moisture, and its biodegradability rate is too high [4,20]. Therefore, most authors blended TPS with other biodegradable polymers such as polylactic acid (PLA) [21,22,23], poly(ε-caprolactone) (PCL) [10,23,24,25], and some others, such as poly (butylene adipate-*co*-terephthalate (PBAT) [7] or polyvinyl alcohol (PVA) [26]. By far the most popular are TPS/PCL blends, which are even produced commercially by at least two big manufacturers: the Novamont company produces TPS/PCL blends named Mater-Bi [27,28] and the Bioplast company produces TPS/PCL blends with trademark Biotec^®^ [29]. Binary TPS/PCL blends aim for technical applications, namely in the field of packaging [30,31,32]. As for biomedical applications, the TPS/PCL particles in solution have been considered as possible microcapsules for drug delivery [33,34], but bulk solid TPS/PCL blends have been mentioned in just two studies: Bou-Francis et al. [24] investigated TPS/PCL together with other biodegradable polymer blends as potential fracture fixation devices (which is suspicious due to the very low modulus of PCL/TPS blends) and Mano et al. [25] studied the thermal properties of TPS, which was blended with various synthetic biodegradable polymers including PCL, and concluded that all starch-based blends *might* be suitable for some medical applications.

This work deals with a reproducible preparation of highly homogeneous TPS/PCL blends for a local release of antibiotics (ATB). The fully biodegradable TPS/PCL/ATB systems are being developed in collaboration with a local hospital for the treatment of strong local infects, such as osteomyelitis [35]. The basic idea has been described in our patent [36]: starch is mixed with ATB easily during our optimized plasticization procedure [16], the resulting TPS/ATB system is melt-mixed with PCL, and the final TPS/PCL/ATB system releases ATB at a rate that is controlled by the blend composition and morphology. More details concerning the biomedical applications of TPS/PCL/ATB systems are summarized in Appendix B, while this work is focused on the optimization of the TPS/PCL preparation procedure, which should yield highly homogeneous TPS/PCL blends with co-continuous morphology and with the minimal thermomechanical degradation of the system. The *perfectly homogeneous material* without local fluctuations is required due to the target application, which requires well-defined ATB release. The *co-continuous morphology* of TPS/PCL blends is necessary for fast ATB release from the TPS phase, as documented in our patent [36]. The *minimal thermomechanical degradation* (i.e., low processing temperatures and low torque forces during starch plasticization and melt-mixing) is extremely important in order to keep the bacteriostatic and bacteriocidic activity of the incorporated antibiotics, as indicated by our parallel experiments (preliminary results for neat TPS/ATB systems have already been published [18]).

## 2. Materials and Methods

### 2.1. Materials

Wheat starch powder (type A) was supplied by Škrobárny Pelhřimov, a.s. (Pelhřimov, Czech Republic). The granular poly(ε-caprolactone) Capa 6800 (PCL; *M* = 80,000 g·mol^−1^, *T_m_* = 58 °C) was supplied by Perstorp Group (Malmö, Sweden). Anhydrous glycerol (>99%), hydrochloric acid (HCl; 35%), and sodium bromide (NaBr; >99%) were purchased from Lach-Ner, s.r.o. (Neratovice, Czech Republic).

### 2.2. TPS/PCL Blend Preparation

The TPS/PCL blends were prepared in two ways. In either case, the first step of our preparation was the plasticization of starch (Section 2.2.1). The second step was either melt-mixing of TPS with PCL (the two-step preparation procedure; Section 2.2.2) or additional melt-mixing and homogenization of TPS followed by the final melt-mixing of TPS and PCL (the three-step preparation procedure; Section 2.2.3). The overall schemes of both preparation procedures are given in the Appendix A. All prepared TPS/PCL samples are listed in Table 1. All prepared samples were stored in a wet desiccator with a saturated solution of sodium bromide (Rh = 57%) between the experiments.

#### 2.2.1. Preparation of TPS by Solution Casting

The solution casted (SC) TPS was prepared according to Ostafińska et al. [16]. The wheat starch (70 wt.%) with glycerol (30 wt.%) and distilled water (6 parts of water per 1 part of starch) were premixed with a magnetic stirrer in a beaker for 30 min at room temperature. The premixed material was transferred to a mechanical stirrer, where it was mixed at elevated temperature (at least 10 min at temperature 65–75 °C until the viscosity significantly increased). The gelatinized starch was spread on a PE foil (in the form of ~1–2 mm thick film) and dried at room conditions (temperature 25 °C, relative humidity ~40%) for 3 days.

#### 2.2.2. Two-Step Preparation of TPS/PCL Blends

In the two-step preparation procedure, the solution casted TPS (step 1) was homogenized with PCL by melt-mixing (MM; step 2) at elevated temperature in the W50EH chamber of the twin-screw laboratory mixer (Brabender Plasti-Corder; Duisburg, Germany). The samples were mixed in the chamber preheated to 120 °C, using rotation speed 60 rpm for 8 min. The blend was compression molded to 2 mm thick plaques in a laboratory hot press (Fontijne Grotnes; Vlaardingen, Netherlands) at 130 °C for 2 min under 50 kN to deaerate, plus another 1 min under 100 kN followed by water cooling for ca 15 min under 100 kN to obtain the final plaques.

#### 2.2.3. Three-Step Preparation of TPS/PCL Blends

In the three-step preparation procedure, the solution casted TPS (step 1) was further homogenized by melt-mixing (step 2) and was then blended with PCL (step 3). The conditions of melt-mixing during steps 2 and 3 (120 °C and 60 rpm for 8 min) were the same as in the two-step procedure described above (Section 2.2.2). The final compression molding conditions were identical to the two-step procedure as well.

### 2.3. TPS/PCL Blend Characterization

#### 2.3.1. Rheology

Rheological properties were measured on a strain-controlled ARES-G2 rheometer (TA Instruments, New Castle, DE, USA) using a parallel plate fixture with a diameter of 30 mm (plates with cross-hatched surface to prevent slipping). Frequency sweep experiments were performed in a frequency range from 0.1 to 100 rad/s at a strain amplitude of 0.05%, and at a constant temperature of 120 °C. The linear viscoelasticity region was determined in view of the dependence of the storage modulus on the strain amplitude, which was measured at 120 °C at a frequency of 1 Hz. The thermal stability of the materials at 120 °C was evaluated in time sweep experiments. To ensure a uniform temperature, all samples were equilibrated for 2 min prior to the start of each type of experiment.

#### 2.3.2. Morphology

The morphology of the TPS/PCL blends was visualized using the scanning electron microscopy (SEM) using a microscope MAIA3 (Tescan, Brno, Czech Republic). All samples were fixed on a metallic support with a conductive silver paste (Leitsilber G302; Christine Groepl, Tulln, Austria), sputter-coated with a thin platinum layer (vacuum sputter coater SCD 050; Leica, Austria; thickness of the Pt layer: approx. 4 nm), and observed by means of secondary electron imaging at accelerating voltage 3 kV.

For each blend composition, we observed two types of specimens: (i) *smoothed and etched surfaces* revealed the phase morphology of the samples and (ii) *fracture surfaces* documented interfacial adhesion between the components. The smooth surfaces were prepared as described elsewhere [35], etched with 6N HCl, washed out with distilled water and dried at room temperature. The etching conditions were adjusted to TPS/PCL blend composition: the blends with the PCL matrix or co-continuous phase (i.e., from TPS/PCL (10/90) to TPS/PCL (70/30) were etched for 10 min in the 6N HCl and the remaining two blends with TPS matrix (i.e., 80/20 and 90/10) were etched in the 6N HCl for only 30 s. The fracture surfaces were prepared through the breaking of the samples submerged in liquid nitrogen [37]. Before the SEM analyses, the samples were placed for 24 h in a dry desiccator (with silica gel), which improved their stability during the vacuum sputter-coating process (minimization of artefact cracks on the surface).

#### 2.3.3. Phase Co-Continuity

Co-continuity of the TPS/PCL (70/30) blend was investigated through laboratory submicron-computed tomography (subμ-CT) using a nano3DX X-ray microscope (Rigaku Corporation, Tokyo, Japan). Part of the sample was cut down to fit in a field of view (FoV) of roughly 0.9 × 0.9 × 0.7 mm^3^. The cut piece was inserted into a Kapton tube and affixed to the walls of the tube using superglue; the tube was then placed on a sample holder and sealed using hot glue. Next, 800 projections of the sample were obtained over a 180° range, using an X-ray generator equipped with a Cu target, with a tube voltage of 40 kV and a current of 30 mA. Exposure per projection was 22 s. A phase retrieval algorithm [38] was applied for contrast enhancement (with a δ/β parameter of 300) and finally an in-house reconstruction algorithm was used to obtain a ~3.4 gigapixel CT volume with a voxel size of 0.528 µm. The volume was processed in VG Studio Max 3.5 (Volume Graphics, Heidelberg, Germany). The hydrogel sample was segmented from its background using ISO50 [39], followed by the manual removal of residual segmented debris and CT artifacts near the edges of the FoV. A region of interest (RoI) was formed around the etched region of the segmented sample and porosity analysis was conducted in this RoI using the “Only threshold” tool of VG Studio.

#### 2.3.4. Micromechanical Properties

Micromechanical properties were obtained from instrumented microindentation hardness testing (MHI) performed with a microindentation combi tester (MCT tester; CSM Instruments, Corcelles, Switzerland). We performed quasi-static measurements with Vickers indenter: a diamond square pyramid with an angle between nonadjacent faces 136° was forced against a smooth sample surface with the following parameters: loading force 981 mN (=100 gf), dwell time (time of maximal load) 60 s, and linear loading and unloading rates 12,000 mN/min (total loading and unloading time ≈ 5 s). The smooth surfaces for MHI testing were prepared with a rotary microtome RM 2155 (Leica, Vienna, Austria). For each sample, at least three independent cut surfaces were prepared and at least 10 indentations were carried out per surface (i.e., each sample was measured at least 30 times) and the final results were averaged. Five micromechanical properties were obtained from *F*–*h* curves (where *F* is the loading force and *h* is the indenter penetration depth) according to ISO 14577-1 standard: indentation modulus (*E_IT_*), indentation hardness (*H_IT_*), Martens hardness (*H_M_*), indentation creep (*C_IT_*), and elastic part of indentation work (*η_IT_*). Figure 1 shows the principle of MHI measurement and evaluation in the case of our TPS/PCL blends; further details can be found in our recent papers [40,41].

#### 2.3.5. Macromechanical Properties

Temperature-dependent dynamic–mechanical properties of the prepared blends were tested on rectangular specimens (40 × 10 × 2 mm) using the same rheometer usedfor the rheological characterization. An oscillatory shear deformation at a constant frequency of 1 Hz and a heating rate of 3 °C/min were applied. The shear deformation was kept in the range from 0.01 to 4% during the temperature sweep, using the AutoStrain function, in order to keep the torque value in the proper range. The evaluated temperature range was from −90 °C to 170 °C.

## 3. Results and Discussion

### 3.1. Relations between Preparation, Rheology and Morphology of TPS/PCL Blends

The main objective of this work was to find a reproducible preparation procedure for highly homogeneous TPS/PCL blends with a co-continuous morphology and minimal thermomechanical degradation of the system during preparation. Blends with these features are required for the release of antibiotics from TPS phase, as explained in the Introduction. To obtain the desired systems, we compared two preparation procedures (Section 3.1.1), considered rheological properties (Section 3.1.2), and characterized the final TPS/PCL morphology by both 2D imaging using a scanning electron microscopy (Section 3.1.3) and 3D imaging using a submicron-computed tomography (Section 3.1.4).

#### 3.1.1. Two-Step vs. Three-Step Preparation Procedure

The advantage of the two-step preparation procedure (Section 2.2.2) is the shorter thermomechanical treatment of TPS (which should contain ATB in the final application), while the three-step preparation procedure (Section 2.2.3) is expected to yield more homogeneous morphology. The first and the last step of both procedures (starch plasticization and the final melt-mixing) were identical. The difference consisted in the intermediate starch homogenization in the three-step procedure. Unfortunately, the melt-mixing of neat TPS exhibited very high torque moments connected with an increase in temperature inside the mixing chamber (from the preset of 120 °C up to 135 °C), while the addition of PCL mitigated these negative effects significantly (as illustrated in Figure 2). This unwanted temperature increase may be decisive, because the two most common antibiotics employed in the treatment of osteomyelitis—vancomycin and gentamicin [35]—are claimed to be stable, just slightly above 120 °C [42,43,44,45]. The decrease in torque moments and in-chamber temperatures during the melt-mixing of TPS/PCL systems as a function of PCL concentration were connected with the much lower viscosity of PCL in comparison with TPS (as discussed in the following Section 3.2.1). In any case, we concluded that the benefit of the three-step preparation (a slightly finer morphology, as illustrated in Figure A1 in Appendix B) did not outweigh the main disadvantage (the higher thermomechanical load of TPS connected with the increased risk of ATB degradation in the final application). Consequently, all following experiments in this study were based on TPS/PCL blends prepared by a two-step preparation procedure.

#### 3.1.2. Rheology of TPS/PCL Blends

Figure 3 summarizes the main output of the oscillatory shear measurements: the frequency dependence and the viscosity (more precisely: of the absolute value of the complex viscosity, |η*|) of all TPS/PCL blends at 120 °C. The results documented two general trends: (i) the viscosity of PCL was much lower than the viscosity of TPS, although the difference somewhat decreased with the increasing frequency and (ii) the addition of PCL to TPS decreased the viscosity of the system, although the situation was more complex at low concentrations of PCL, as discussed below.

For the blend compositions close to the neat components, i.e., for the blends TPS/PCL (100/0 and 90/10) and TPS/PCL (0/100, 10/90, and 20/80), the complex viscosity curves in Figure 3 stayed close to those characteristic of the neat constituent phases. The slight increase in the blend viscosity with respect to the neat constituent is typical of the particulate blends with good interfacial adhesion, as summarized elsewhere [46]. The highest viscosity was observed for the TPS/PCL (80/20) blend with high-viscosity matrix and PCL minority phase. The blends with an even higher content of PCL, i.e., the blends from TPS/PCL (70/30) to TPS/PCL (30/70), exhibited a gradual decrease in viscosity, which indicated the formation of the continuous structure of PCL. This was in perfect agreement with the results of SEM (Section 3.1.3.) and subμ-CT (3.1.4), which confirmed that the blends with TPS content ≥30 wt.% contained a continuous PCL phase. The steepest viscosity decrease was observed for the blends with 30 and 40 wt.% of TPS. The comparison between the values of storage and the loss moduli (Appendix C, Figure A2) suggested that the blends with TPS content above 30 wt.% behaved like physical gels (*G*′ > *G*″), while the blends with TPS content below 30 wt.% exhibited liquid-like behavior (*G*′ < *G*″) at a given temperature and experimental conditions [47].

The good interfacial adhesion between TPS and PCL, suggested by the rheological experiments, was confirmed by SEM micrographs that showed the fracture surfaces of TPS/PCL blends (Appendix C, Figure A3). The good adhesion could be attributed to the affinity of both phases to each other (hydrogen bridging between hydroxyl groups of starch and carboxyl groups of PCL). The absolute value of complex viscosity (η*), in oscillatory shear experiments at a given frequency (ω), is proportional to the absolute value of complex modulus (|*G**| = ω ∙ |η*|; [47]), which represents the overall resistance of the investigated system to deformation. The high resistance to the deformation of the TPS/PCL blends with 20–40 wt.% of PCL (manifested by high |η*|) could only be observed in oscillatory experiments, where the applied deformations were small (within the linear viscoelastic region). During the real melt-mixing of the blends, the shear forces were higher, the chemical bridges between both phases were broken, and the lower-viscosity PCL component acted as a lubricant, which decreased the torque moments and processing temperatures in the whole concentration range, in agreement with the experimental observations described above (Section 3.1.1). 

#### 3.1.3. Morphology of TPS/PCL Blends

Morphology of TPS/PCL blends was visualized by SEM microscopy (Figure 4). The blends with higher concentrations of PCL (Figure 4a–e) contained TPS particles in the PCL matrix. The size of the TPS particles *decreased* with the *increasing* concentration of TPS component, although most blends exhibited the opposite trend—bigger particles at higher concentrations of the minority components [48,49,50]. This nontypical feature of TPS/PCL systems resulted from the much higher viscosity of TPS in comparison with PCL at the processing temperature of 120 °C (as documented in Figure 3).

At low TPS concentrations, the highly viscous and compact TPS particles could not be fragmented in the low-viscosity PCL matrix. At higher TPS concentration, the overall viscosity of the system increased (Figure 3), which resulted in higher shear forces during melt-mixing that caused a more frequent particle breakup and the structure became finer. The blends with higher concentration of TPS showed a co-continuous morphology (Figure 4f,g) and, finally, the reversed morphology of the TPS matrix containing PCL particles (Figure 4h,i). The reason why the co-continuous morphology region of the TPS/PCL blends was shifted from 50/50 concentration and the proof of the TPS phase co-continuity of TPS/PCL (70/30) blend are given in the Section 3.1.4.

#### 3.1.4. Phase Co-Continuity of TPS/PCL Blends

The theoretical prediction and experimental verification of the phase co-continuity in TPS/PCL blends are shown in Figure 5. The prediction of phase co-continuity was based on the classical analysis of Paul and Barlow [51], which we summarize in the form of a simple scheme (Figure 5a). Briefly, point (1) in Figure 5a shows that the co-continuous morphology of a binary polymer blend occurs at 50/50 composition (volume fraction ratio = 1) on the condition that the viscosities of the individual components are equal (viscosity ratio = 1). Nevertheless, in our case, the TPS/PCL viscosity ratio was significantly higher than 1 (point (2) in Figure 5a), and, consequently, the phase co-continuity was observed at higher TPS/PCL volume ratios of 60/40 and 70/30 (Figure 4 and point (3) in Figure 5a). The proof of phase co-continuity was obtained by submicron-computed tomography (subμ-CT; Figure 5b). Analysis of subμ-CT data showed that the etching of the TPS/PCL (70/30) blend, for which the two-dimensional SEM micrographs (Figure 4) indicated the highest degree of co-continuity, removed the TPS phase up to a depth of about 100 µm. Porosity analysis revealed that the etched volume forms a single interconnected network spanning the entire width and depth of the etched region. This network makes up about 95.34% of the total porosity identified in the RoI, confirming the hypothesized high degree of co-continuity in the sample. The true degree of co-continuity may be even higher when accounting for pores at the edges of the RoI and pores connected by channels which could not be resolved at the spatial resolution of the CT measurement.

### 3.2. Micro- and Macromechanical Properties of TPS/PCL Blends

The second objective of this work was the characterization of the mechanical behavior of TPS/PCL blends in order to verify whether the soft TPS phase was stiffened by the addition of PCL. Moreover, we wanted to describe the mechanical performance of all blends in both microscale (using MHI; Section 3.2.1) and macroscale (using DMTA; Section 3.2.2). The microscale characterization would be advantageous for future analyses of TPS/PCL/ATB systems, which are usually prepared in low amounts due to costly antibiotics. Therefore, we quantified the correlations between the corresponding properties from MHI and DMTA measurements to check if the microscale characterization yields relevant results (Section 3.2.3).

#### 3.2.1. Micromechanical Properties

Instrumented microindentation hardness testing (MHI) yielded five micromechanical properties of TPS/PCL blends: indentation modulus (*E_IT_*; proportional to macroscopic moduli), indentation hardness (*H_IT_*; proportional to macroscopic yield stress), indentation creep (*C_IT_*; related to macroscopic creep), elastic part of the indentation work (*η_IT_*; connected with overall elasticity of the sample), and Martens hardness (*H_M_*; analogous to *H_IT_*, but calculated independently on the Oliver and Pharr theory [52]). Figure 6 shows the first four micromechanical properties (*E_IT_*, *H_IT_*, *C_IT_*, and *η_IT_*) plotted as a function of the TPS/PCL composition and compared with two predictive models (linear model, LIN, and equivalent box model, EBM). The LIN model, which is also referred as rule-of-mixing or additivity law, assumes that the arbitrary property *P* of a multicomponent system is a linear combination of its component properties:
(1)$$P=\sum_{i}v_{i}P_{i}$$
where *P_i_* and *v_i_* represent a given property and volume fraction of *i*-th component, respectively. The simple LIN model holds surprisingly well in some cases, such as the elastic moduli of polymer composites with infinite fibers [50,53] or microhardness of semicrystalline polymers [54,55]. Nevertheless, for polymer blends, the LIN model is just a rough approximation, representing the upper achievable limit of the real blend properties [55,56]. The more advanced EBM model, which describes the selected properties of isotropic binary blends, also takes into account the degree of phase co-continuity and interfacial adhesion [57]. In terms of EBM, the indentation modulus (*E_IT_*) and hardness (*H_IT_*) of a binary polymer blend are defined by Equations (2) and (3), respectively:(2)$$E_{IT}=E_{1}v_{1p}+E_{2}v_{2p}+v_{s}^{2}/\left[\left(v_{1s}/E_{1}\right)+\left(v_{2s}/E_{2}\right)\right]$$
(3)$$H_{IT}=H_{1}v_{1p}+H_{2}v_{2p}+AH_{1}v_{s}$$
where *E_i_* and *H_i_* are the component properties and *v_ij_* represents volume fractions (subscript (*i*) identifies the component and subscript (*j*) determines the volume of the component in the parallel (*p*) and serial (*s*) branch of the EBM model, which correspond to volume fractions with continuous and particulate morphology, respectively). The parameter *A* represents interfacial adhesion (the values *A* = 0 and 1 mean negligible and perfect adhesion, respectively). The details of the EBM model were described in the original work of Kolarik [57] and have also been summarized in our recent studies [50,58]. It is worth noting that the EBM model was derived for macroscopic elastic modulus (*E*) and yield stress (*Y*), but it has been demonstrated that EBM also predicted micromechanical properties *E_IT_* ≈ *E* [52] and *H_IT_* ≈ 3 *Y* [59]. The volume fractions (*v_ij_*) in the EBM model can be either estimated from additional experiments (for example, from SEM micrographs [48]) or calculated theoretically (from percolation theory [57]), while the adhesion parameter (*A*) is obtained from the fitting of experimental data (hardness or yield stress as a function of composition; Equation (3), ref. [48,60]).

In this work, the EBM predictions for *E_IT_* and *H_IT_* (Figure 6A,B) were based on theoretically calculated *v_ij_* values (which were shown to be a good first approximation in many binary polymer blends [48,57,58,60]) and maximal interfacial adhesion (*A* = 1). The fact that both *E_IT_* and *H_IT_* exceeded the theoretical EBM predictions evidenced strong interfacial adhesion and very good compatibility between TPS and PCL, which was is in agreement with the rheological and morphological results (Section 3.1.2 and Section 3.1.3). The two supplementary properties, *C_IT_* and *η_IT_* (Figure 6C,D), were in reasonable agreement with LIN predictions, which reconfirmed the good compatibility of TPS and PCL polymers, because LIN model represents the general upper achievable limit of blend properties, as discussed above.

#### 3.2.2. Macromechanical Properties

The macroscale thermomechanical properties of the TPS/PCL blends were analyzed by oscillatory torsion experiments (DMTA). The DMTA results are shown in Figure 7 in the form of storage moduli (*G*′) as a function of temperature (*G*′ = f (*T*) curves). The complete set of DMTA properties (storage, loss, and complex moduli, together with damping factors) in a temperature range from −80 °C to 160 °C is shown in Appendix D (Figure A6).

In the high temperature region (from ~60 °C to 160 °C), the blends with the TPS content ≥30 wt.% behaved like elastic gels (approximately constant values of *G*′ above 60 °C), while the blends with the TPS content ≤20 wt.% behaved like polymer melts (a steep decrease of *G*′ above 60 °C). This agreed very well with the rheological measurements performed at 120 °C (Section 3.1.2), which proved that the TPS/PCL blends with TPS ≥ 30 wt.% exhibited a mostly solid-like character (*G*′ > *G*″ in Figure A3), whereas TPS/PCL blends with lower TPS content showed a liquid-like character (*G*″ > *G*′ in Figure A3). As the temperature of 120 °C was approximately equal to the processing temperature (Figure 2), the above-discussed DMTA results were also connected with the TPS/PCL behavior during melt-mixing. The strong decrease of *G*′ around 60 °C corresponded to the melting temperature of PCL.

In the medium temperature region (from ca 5 °C to 60 °C), the storage moduli of TPS/PCL blends increased monotonously with the PCL content. This was connected with the fact that TPS started to soften above −10 °C, while PCL softened by melting above 60 °C (and so it was stiffer than TPS at medium temperatures). The changes in shape of the DMTA curves corresponded approximately to the weighted sum of the blend components, which is typical for immiscible blends. The inset in Figure 7 shows the *G*′ values around the room temperature of 25 °C, where the storage modulus decreased by 1 order of magnitude if going from neat PCL (ductile semicrystalline polymer above *T_m_* with *G*′ ~150 MPa) to neat TPS (soft rubber with *G*′ ~16 MPa). The medium temperature region and especially the temperatures around 25 °C were important for the correlations between the results of macroscale DMTA testing and microscale MHI testing (which are discussed in the Section 3.2.3) because the MHI measurements were performed at room temperature. 

In the low temperature region (temperatures below 5 °C), both blend components exhibited glass transition temperatures (*T_g_*) and the behavior of the TPS/PCL systems changed accordingly. Neat PCL showed *T_g_* at −55 °C (a step of the *G*′ = f(*T*) curve; an even larger step was observed at the PCL melting temperature around 60 °C). Neat TPS showed two *T_g_* temperatures, corresponding to two steps of the *G*′ = f(*T*) curve: the first step was around −57 °C (overlapping with the *T_g_* of PCL) and the second was a very broad transition at around 0 °C (the onset of the step in Figure 7 at −10 °C and the end of the step at 40 °C). In accordance with the previous studies [61,62], the lower *T_g_* of TPS was assigned to the *glycerol-rich phase*, while the higher *T_g_* was assigned to the *starch-rich phase*. As expected, the glass transition temperatures could also be observed in the form of peaks at G″ = f(*T*) and tan(δ) = f(*T*) curves (Appendix D, Figure A6). As for TPS/PCL blends, the rising TPS content led to higher moduli in the fully vitrified state below −80 °C, to higher moduli in the rubbery plateau above 60 °C, to a diminishing melting step of PCL at 60 °C, and to a more prominent glass transition of TPS observed near 2 °C (as documented in both Figure 7 and Figure A6 in Appendix D).

#### 3.2.3. Correlations between Micro- and Micromechanical Properties

The mechanical properties measured in this work can be divided into two groups: stiffness-related properties (*G*′, *E_IT_*, *H_IT_*, and *H_M_*) and viscosity-related properties (*G*″, tan(δ), *C_IT_*, and *η_IT_*). The same division of mechanical properties was employed and justified in our previous work [40]. The stiffness-related properties were more important here because the soft TPS matrix was expected to harden thanks to the PCL addition. As noted in the previous paragraph, all correlations between micro- and macromechanical properties were evaluated for the room temperature of 25 °C.

The correlations between all stiffness-related properties were very strong, as evidenced in Figure 8. The fact that the macromechanical storage modulus (*G*′) was strongly correlated with all stiffness-related micromechanical properties (*E_IT_*, *H_IT_* and *H_M_*) proved that the micromechanical characterization of the TPS/PCL blends was reliable and suitable for ongoing parallel studies with lower volumes of the blends containing antibiotics. Moreover, the strong correlation between the first two micromechanical properties (*E_IT_* and *H_IT_*) evaluated in terms of the Oliver and Pharr theory (O & P theory; [52]) and the last property (Martens hardness, *H_M_*), which is evaluated directly from the experimental data [63], evidenced that the O & P theory is a good approximation in the case of TPS/PCL blends. The same trend of *H_IT_* and *H_M_* dependencies on the blend composition is also evident in Figure A5 from Appendix D.

The correlations between viscosity-related properties were somewhat weaker, but identifiable. Microscale properties (*C_IT_* and *η_IT_*) showed quite strong and logical trends, as illustrated in Figure 6: the increasing concentration of the soft and viscous TPS component resulted in lower creep resistance (increasing *C_IT_*; Figure 6C) and lower elasticity (decreasing *η_IT_*; Figure 6D). Macroscale properties related to viscosity (*G*″ and tan(δ)) changed with the increasing concentration of TPS in two different ways, as documented in Figure A5: the values of *G*″ at 25 °C were similar for all TPS/PCL blends (Figure A5B) without a strong trend, but the values of tan(δ) at 25 °C increased clearly and monotonously with the increasing TPS concentration (Figure A5D). Considering that the shear moduli *G*′, *G*″, and |*G**| represent the deformation energy stored due to elasticity, deformation energy lost due to viscous flow, and total material resistance to deformation, respectively, while the damping factor tan(δ) = *G*″/*G*′ represents ratio of the viscous and elastic portion of the deformation behavior [47], the final interpretation is as follows: For increasing the concentration of TPS in the TPS/PCL blends at 25 °C, the overall elasticity of the system decreased (Figure A5A), the overall viscosity was roughly constant (Figure A5B), the total resistance to deformation decreased as it was dominated by the elastic contribution *G*′ (similar trends in Figure A5A,C), and the viscous behavior played an increasingly important role (Figure A5D), which was quite reasonable as the TPS represented the softer and less elastic component of the blend. All correlations are summarized in Appendix D in the form of a heatmap (Figure A7).

## 4. Conclusions

We have described a reproducible preparation of highly homogeneous TPS/PCL blends and characterized their properties in both macro- and microscale. The biodegradable TPS/PCL systems, containing antibiotics in TPS phase, are being developed for the treatment of strong infects within parallel collaboration with a local hospital. This work was focused on the optimization of the TPS/PCL preparation procedure, which should yield perfectly homogeneous TPS/PCL blends with a co-continuous morphology and minimal thermomechanical degradation of the system. The *perfectly homogeneous material* is necessary for reproducible and well-defined ATB release. The *co-continuous morphology* of TPS/PCL blends is required for fast ATB release from the fast-degrading TPS phase. The *minimal thermomechanical degradation* during the preparation procedure is essential in order to keep the high activity of the antibiotics in the final application. The main conclusions of the current study can be summarized as follows:1.The two-step preparation procedure, comprising the preparation of TPS by solution casting and melt-mixing of TPS with PCL, yielded highly homogeneous TPS/PCL blends reliably and reproducibly;2.The TPS/PCL (70/30) blends exhibited co-continuous morphology and the processing temperature during the preparation should be kept below 130 °C due to the addition of the less viscous PCL phase;3.The TPS/PCL blends showed high interfacial adhesion and their stiffness-related properties (elastic moduli and hardness) were even slightly above the theoretical predictions based on the EBM model. In other words, the very soft TPS matrix could be stiffened quite efficiently by means of the compatible PCL component;4.The mechanical properties of the TPS/PCL blends were characterized in both macroscale (by DMTA) and microscale (by MHI). The correlations between corresponding macro- and micromechanical properties were strong. This confirmed that micromechanical measurements were quite sufficient for the characterization of the mechanical performance of the TPS/PCL blends. Consequently, the future mechanical characterizations of the TPS/PCL/ATB systems, which are usually available in lower volumes due to costly antibiotics, could be performed simply by microindentation hardness testing, which does not require large specimens.

## Figures and Tables

**Figure 1 materials-15-01101-f001:**
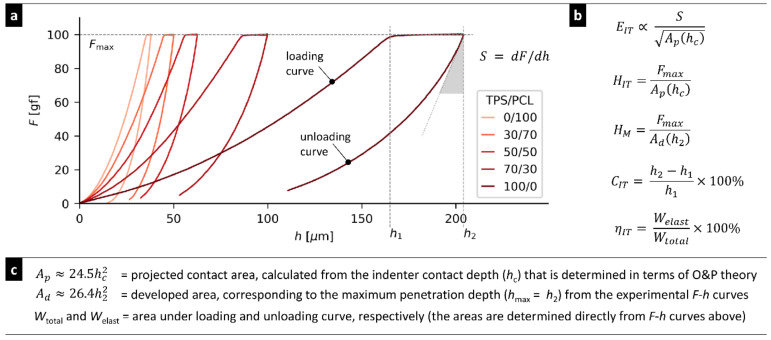
Principle of MHI measurements, which were employed in this work: (**a**) Micromechanical properties were deduced from the experimental *F*–*h* curves (where *F* is the indenter loading force and *h* is the indenter penetration depth) by means of (**b**) formulas and (**c**) relations containing experimental parameters, such as maximum loading force (*F_max_*), slope at the beginning of the unloading curve (*S*), penetration depths at the beginning and end of the maximal load (*h*_1_ and *h*_2_), and areas under loading and unloading curve (*W_elast_* and *W_total_*). Additional parameter, contact depth (*h_c_*), was calculated in terms of the Oliver and Pharr theory and employed in the calculation of *E_IT_* and *H_IT_*, as described in our previous work [41,42]. The figure shows real, representative *F*–*h* curves of TPS/PCL systems, which the illustrate substantial changes of all studied properties as a function of composition.

**Figure 2 materials-15-01101-f002:**
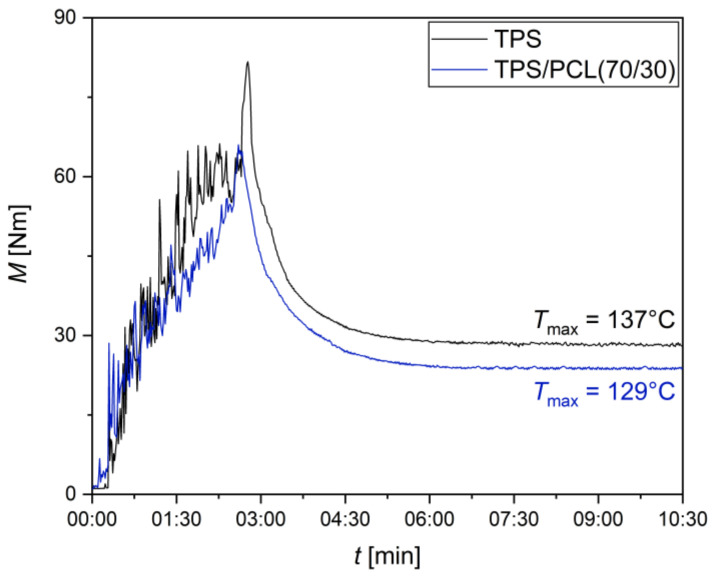
Torque moments as a function of melt-mixing time and maximum temperatures during melt-mixing of neat TPS (black line) and TPS/PCL (70/30) blend (blue line). The initial increase of the torque moments corresponds to the filling of the mixing chamber, which was heated to 120 °C.

**Figure 3 materials-15-01101-f003:**
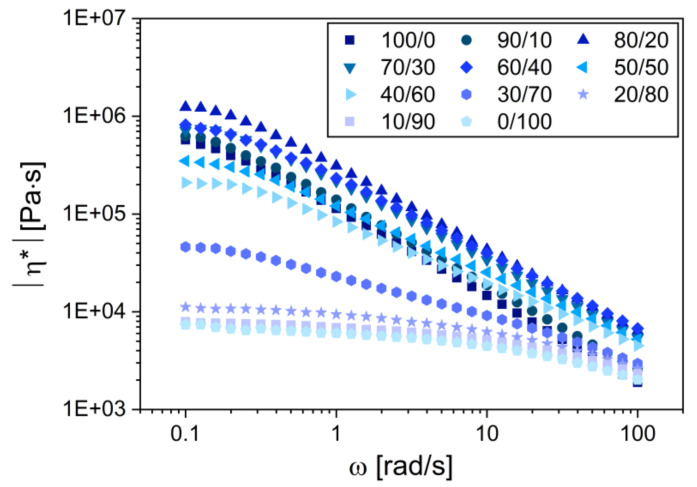
Absolute value of the complex viscosity (|η*|) as a function of oscillatory shear angular frequency (ω) during the oscillatory shear measurements for all TPS/PCL blends.

**Figure 4 materials-15-01101-f004:**
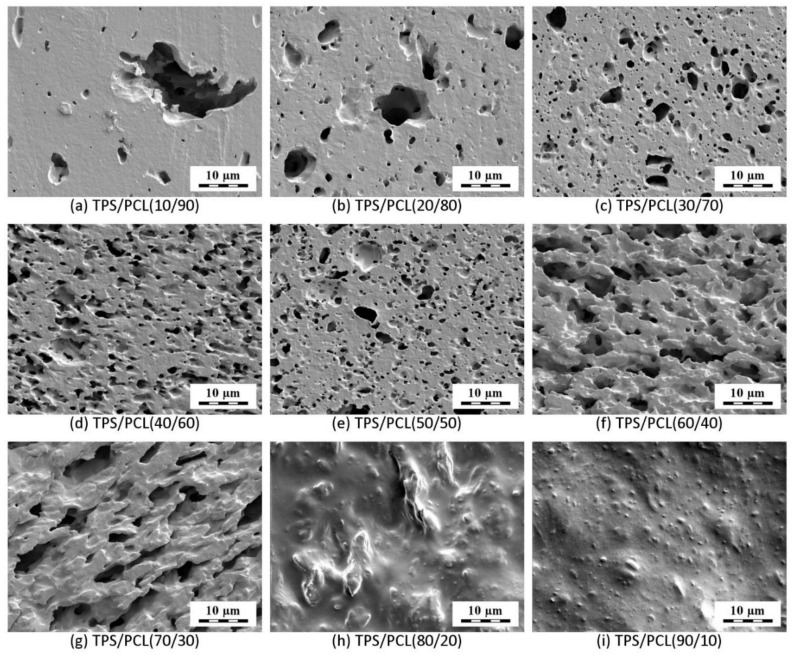
SEM micrographs showing smoothed surfaces of the TPS/PCL blends; the TPS phase was etched off by hydrochloric acid.

**Figure 5 materials-15-01101-f005:**
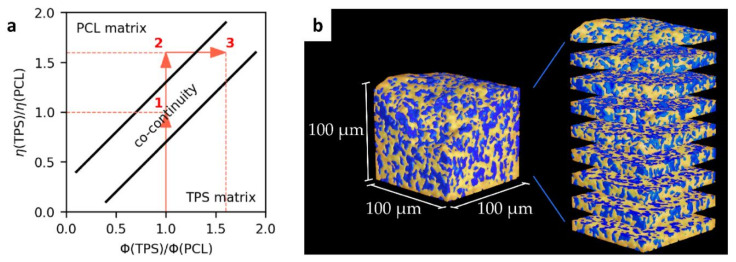
Phase co-continuity in TPS/PCL blends: (**a**) a theoretical prediction of the TPS/PCL composition exhibiting co-continuous morphology and (**b**) an experimental proof of co-continuity in the TPS/PCL (70/30) blend by means of subμ-CT measurement. The theoretical prediction (**a**) suggested that co-continuous morphology should appear at higher concentrations of TPS (as explained in Section 3.1.4). The experimental verification of the co-continuity of the TPS/PCL (70/30) blend (**b**) is displayed in the form of a 3D reconstruction of 100 × 100 × 100 μm^3^ volume of the sample measured by subμ-CT. The model illustrates co-continuity in both phases. The etched-off phase (TPS; shown in blue) forms interconnected pores. The connections among the pores can be observed in the thin slices on the right.

**Figure 6 materials-15-01101-f006:**
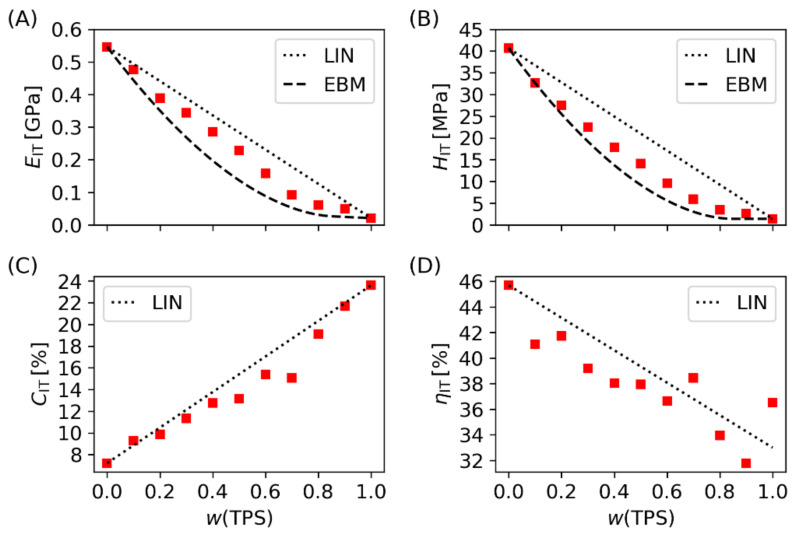
Micromechanical properties of TPS, PCL, and of the TPS/PCL blends (red points) vs. theoretical predictions (black lines). The plots show (**A**) indentation hardness, (**B**) indentation modulus, (**C**) indentation creep and (**D**) elastic work of indentation. The theoretical predictions comprise linear model (LIN; the final blend property is a linear combination of the blend component properties) and equivalent box model (EBM; a simple predictive scheme for isotropic polymer blends, which considers phase continuity and interfacial adhesion). The models are described in Section 3.2.1.

**Figure 7 materials-15-01101-f007:**
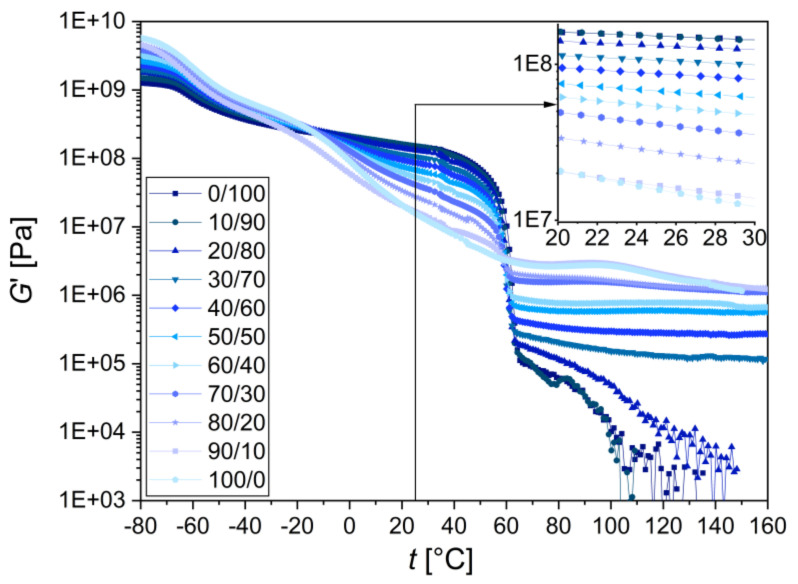
Results of DMTA: storage modulus (*G*′) is a function of temperature (*T*) for all TPS/PCL blends. The inset shows detail of the region around room temperature.

**Figure 8 materials-15-01101-f008:**
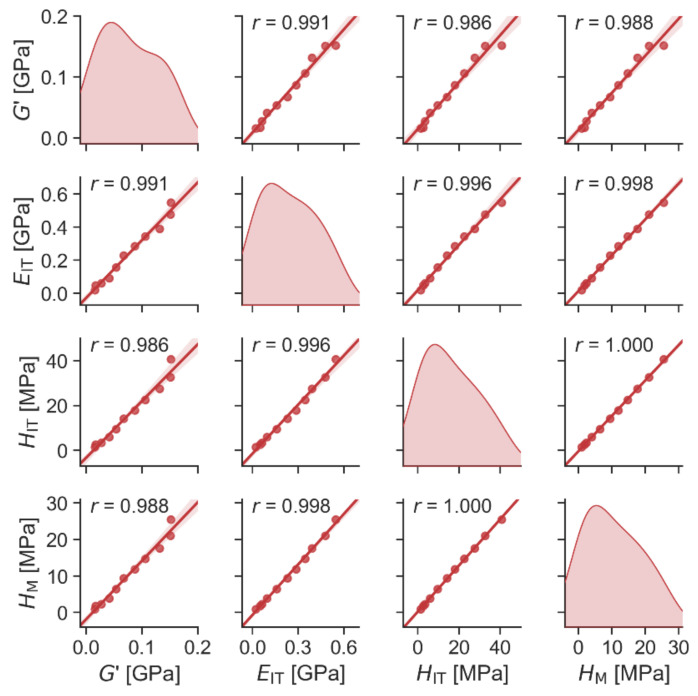
Scatterplot matrix graph showing correlation between selected micromechanical (*G*′) and micromechanical properties (*E_IT_*, *H_IT_* and *H_M_*). Diagonal elements of the scatterplot matrix graph show distribution of the measured quantities, whereas off-diagonal elements show correlations between each pair of quantities. The translucent bands around the regression lines represent 95% confidence interval of the regression estimate. Moreover, all off-diagonal plots show the values of Pearson’s correlation coefficient *r* in the upper right corner.

**Table 1 materials-15-01101-t001:** List of the prepared TPS/PCL blends; each blend was prepared by both two-step procedure (Section 2.2.2) and three-step procedure (Section 2.2.3).

Sample ID	Wt.% (TPS)	Wt.% (PCL)
TPS/PCL (0/100)	0	100
TPS/PCL (10/90)	10	90
TPS/PCL (20/80)	20	80
TPS/PCL (30/70)	30	70
TPS/PCL (40/60)	40	60
TPS/PCL (50/50)	50	50
TPS/PCL (60/40)	60	40
TPS/PCL (70/30)	70	30
TPS/PCL (80/20)	80	20
TPS/PCL (90/10)	90	10
TPS/PCL (100/0)	100	0

## Data Availability

Data are available at request to the corresponding author.

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
