# Peer review of "Biodegradable Thermoplastic Starch/Polycaprolactone Blends with Co-Continuous Morphology Suitable for Local Release of Antibiotics"

_materials, 2022, doi:10.3390/ma15031101_

Round 1

Reviewer 1 Report

The present manuscript entitled "Biodegradable thermoplastic starch/polycaprolactone blends with co-continuous morphology suitable for local release of antibiotics" is a well articulated research manuscript.

The authors have described the physical properties of the homogeneous thermoplastic starch/pol(ε-caprolactone) blends (TPS/PCL), however it will be useful to show that the antibiotic release is still biologically active . This can be done using a simple agar plate assay. I would recommend the inclusion of this assay (Kirby-Bauer Disk Diffusion test)

Minor point:

Figure 7 is bit difficult to read I would recommend to make this in two figures or expand it to make it easier for the reader to understand.

Author Response

Point-by-point answers to comments of reviewer #1

Comment: The present manuscript entitled "Biodegradable thermoplastic starch/polycaprolactone blends with co-continuous morphology suitable for local release of antibiotics" is a well-articulated research manuscript.

Answer: We thank the reviewer for his/her positive evaluation.

Comment: The authors have described the physical properties of the homogeneous thermoplastic starch/pol(ε-caprolactone) blends (TPS/PCL), however it will be useful to show that the antibiotic release is still biologically active. This can be done using a simple agar plate assay. I would recommend the inclusion of this assay (Kirby-Bauer Disk Diffusion test)

Answer: In the introduction we explain that our work is focused on reproducible preparation of highly homogeneous TPS/PCL blends and their morphological, rheological and mechanical characterization, while biomedical applications were just briefly mentioned as the final application, which was described in our parallel patent (reference [36] in the manuscript). Nevertheless, we added some results (standard microbiological tube dilution tests) evidencing that the ATB in TPS/PCL systems keeps its biological activity. These results were inserted in new Appendix A (Figure A1). The reference to Appendix A is given in the last paragraph of Introduction.

Minor point:

Figure 7 is bit difficult to read I would recommend to make this in two figures or expand it to make it easier for the reader to understand.

Answer: The figure was slightly modified (larger font within the inset) so that it was easier to read.

Reviewer 2 Report

In this paper, the authors developed a reproducible preparation method for biodegradable thermoplastic starch/polycaprolactone blends exhibiting co-continuous morphology potentially targeted for local release of antibiotics. The work looks nice and reliable. However, the authors are claiming their develop product suitable for local release of antibiotics but they didn't perform any study to justify their claim. It would be better if authors provide some data regarding the efficiency of local release of antibiotics. Overall, I recommend this article to be published in Materials after adding suggested data.

Author Response

Point-by-point answers to comments of reviewer #2

Comment: In this paper, the authors developed a reproducible preparation method for biodegradable thermoplastic starch/polycaprolactone blends exhibiting co-continuous morphology potentially targeted for local release of antibiotics. The work looks nice and reliable. However, the authors are claiming their develop product suitable for local release of antibiotics but they didn't perform any study to justify their claim. It would be better if authors provide some data regarding the efficiency of local release of antibiotics. Overall, I recommend this article to be published in Materials after adding suggested data.

Answer: We thank the reviewer for his/her positive evaluation. We added the required data concerning local release of antibiotics to a newly inserted Appendix A. The reference to Appendix A is given in the last paragraph of Introduction.

Reviewer 3 Report

In this work, the authors reported the reproducible preparation of biodegradable TPS/PCL blends. And their thermomechanical properties were substantially characterized. Overall, I believe this work is fitting to Materials after several major issues listed below are properly addressed.

  1. The authors claimed this TPS/PCL blend might be suitable for local release of antibiotics, which should be supported by solid experimental data.
  2. How about the superiority of TPS/PCL in this work compared with commercialized products or other reported analogs?
  3. In the introduction section, the necessity to develop TPS/PCL blend for release of antibiotics should be elaborated. For instance, why to use biodegradable polymer for release of antibiotics.
  4. Biodegradable property of TPS/PCL blend should be studied.
  5. The biomedical purpose for this TPS/PCL/ATB is quite confusing. The authors mentioned that the fully biodegradable TPS/PCL/ATB systems are being developed in collaboration with local hospital for treatment for strong local infects. More details should be described for this biomedical application. For instance, if the blend is to be used as coating materials or something else?
  6. The authors should better draw a scheme for TPS/PCL blend and also its preparation process. It would be much easier for readers to get the point.

Author Response

Point-by-point answers to comments of reviewer #3

Comment: In this work, the authors reported the reproducible preparation of biodegradable TPS/PCL blends. And their thermomechanical properties were substantially characterized. Overall, I believe this work is fitting to Materials after several major issues listed below are properly addressed.

Answer: We thank the reviewer for his/her overall positive evaluation and comments that helped us to clarify some issues concerning potential applications of TPS/PCL blends. Most of the comments of the third reviewer (which are listed and answered below), were connected with possible biomedical applications of our systems. As we explained in the Introduction, this study was focused on reproducible preparation of highly homogeneous TPS/PCL blends and their morphological, rheological and mechanical characterization, while biomedical applications were just briefly mentioned as the final application, which was described in our parallel patent (reference [36] in the manuscript). Nevertheless, the reviewer is right that some questions concerning the final application should be discussed in a bit more detail. Therefore, we decided to insert Appendix A, which should answer these questions (and analogous questions of the other reviewers). The reference to Appendix A is given in the last paragraph of Introduction section of the revised manuscript.

(1) The authors claimed this TPS/PCL blend might be suitable for local release of antibiotics, which should be supported by solid experimental data.

Answer: This question is answered in item (2) of the newly inserted Appendix A, which reads:

The TPS/PCL/ATB systems (where ATB = suitable antibiotic dispersed in TPS phase) can be employed in the treatment of strong local infects such as osteomyelitis, which is an infection of bone tissue. The two major antibiotics for osteomyelitis treatment are vancomycin and gentamicin [35]. Both substances exhibit reasonable resistance to thermal degradation [42–45] and, as a result, they can survive the TPS/PCL preparation procedure and keep their activity, as we have already documented by standard microbiological tube-dilution tests [18]. The tests were performed in our initial study about simplified TPS/ATB systems [18] and also in our parallel experiments with TPS/PCL/ATB systems (unpublished results, illustrated in Figure A1 below).

The newly inserted Figure A1 shows our preliminary, not-yet-published results of tube-dilution tests performed on TPS/PCL/ATB systems.

(2) How about the superiority of TPS/PCL in this work compared with commercialized products or other reported analogs?

Answer: This question is answered in item (3) of the newly inserted Appendix A, which reads:

In the case of above-discussed bone infections, the commercial carrier systems for lo-cal release of antibiotics comprise: (i) bone cement (basically poly(methyl methacrylate)), (ii) capsules of highly porous plaster (CaSO4 × 2H2O) or (iii) fibrin-based foams [36]. All three systems can contain and release ATB locally, but their mechanical properties are not optimal: bone cement is stiff polymer deep below its glass transition temperature, highly porous plaster is hard and brittle inorganic material, and fibrin foams are too soft. Moreover, ATB release rate for each of the above-listed materials is constant. In contrast, our TPS/PCL system is a ductile plastic material with tunable mechanical properties, cuttable with scissors at room temperature, shapeable upon heating at 80 °C, and ATB release can be controlled by its composition and morphology. Consequently, TPS/PLC blends are suitable not only as the ATB releasing spacers in bone infections and defects (which was our originally intended application), but also in other biomedical fields, such as sticking plasters, adhesive bandages and patches for wound healing.

(3) In the introduction section, the necessity to develop TPS/PCL blend for release of antibiotics should be elaborated. For instance, why to use biodegradable polymer for release of antibiotics.

Answer: In the introduction section, we inserted a reference to Appendix A, which answers the questions above and summarizes also the potential advantages of TPS/PCL blends in comparison with existing commercial systems for the specific case of osteomyelitis treatment and some other applications – see also the answer to the previous comment.

(4) Biodegradable property of TPS/PCL blend should be studied.

Answer: References to biodegradability of TPS and PCL are summarized in item (1) newly inserted Appendix A, which reads:

Both TPS and PCL are biocompatible and biodegradable polymers. The biocompatibility of both polymers is well documented [1,64]. The biodegradability rate of TPS/PCL blends in vivo is given by PCL, because the TPS component degrades much faster (within a few days) than PCL (ca 2–4 years) [65,66,67].

(5) The biomedical purpose for this TPS/PCL/ATB is quite confusing. The authors mentioned that the fully biodegradable TPS/PCL/ATB systems are being developed in collaboration with local hospital for treatment for strong local infects. More details should be described for this biomedical application. For instance, if the blend is to be used as coating materials or something else?

Answer: This comment has already been answered above – see the response to comment #2 – the applications are listed in item (3) of the newly inserted Appendix A.

(6) The authors should better draw a scheme for TPS/PCL blend and also its preparation process. It would be much easier for readers to get the point.

Answer: The reviewer is right, such a scheme may be useful for possible readers that are not familiar with the starch plasticization etc. Therefore, a simple scheme was drawn, added to Supplementary Information and the reference to the scheme in SI was given in the Experimental section (section 2.2.) of the revised manuscript.

Round 2

Reviewer 1 Report

I am satisfied with the revised submission, and would recommend accepting the manuscript for publication 

Reviewer 2 Report

The authors have made the needed changes in the revised manuscript t. I am pleased to recommend this manuscript to be published in Materials.

Reviewer 3 Report

The authors answered my questions and I have no other questions.